# Update on Renal Cell Carcinoma Diagnosis with Novel Imaging Approaches

**DOI:** 10.3390/cancers16101926

**Published:** 2024-05-18

**Authors:** Marie-France Bellin, Catarina Valente, Omar Bekdache, Florian Maxwell, Cristina Balasa, Alexia Savignac, Olivier Meyrignac

**Affiliations:** 1Service de Radiologie Diagnostique et Interventionnelle, Hôpital de Bicêtre AP-HP, 78 Rue du Général Leclerc, 94275 Le Kremlin-Bicêtre, France; catarina.valente@aphp.fr (C.V.); omar.bekdache@aphp.fr (O.B.); florian.maxwell@aphp.fr (F.M.); alexia.savignac@aphp.fr (A.S.); olivier.meyrignac@aphp.fr (O.M.); 2Faculté de Médecine, University of Paris-Saclay, 63 Rue Gabriel Péri, 94276 Le Kremlin-Bicêtre, France; 3BioMaps, UMR1281 INSERM, CEA, CNRS, University of Paris-Saclay, 94805 Villejuif, France

**Keywords:** renal cell carcinoma, dual-energy CT, spectral CT, photon-counting detector CT, quantitative computed tomography, multiparametric MRI, contrast-enhanced ultrasound, sestamibi SPECT/CT, PSMA PET/CT, radiomics, artificial intelligence, Bosniak classification version 2019, clear cell likelihood score, AUA guidelines

## Abstract

**Simple Summary:**

The incidence of renal cell carcinoma (RCC) is increasing due to the expansion of cross-sectional imaging and advanced imaging techniques. They allow for the detection of tumors at an earlier stage, but there are often overlapping similarities in the appearance of benign and malignant renal tumors. This review presents and discusses the ever-evolving landscape of imaging techniques that can be used to detect and diagnose renal cell carcinoma, including its major histologic subtypes. It also provides insight into recently proposed or updated imaging algorithms and guidelines for the diagnosis of RCC. The review considers the major advances in spectral CT, photo- counting CT, multiparametric MRI, contrast-enhanced ultrasound, sestamibi SPECT/CT, PSMA PET/CT, radiomics, artificial intelligence, Bosniak classification version 2019, clear cell likelihood score, and AUA guidelines. The goal for radiologists is to be better equipped to guide the diagnosis and management of these patients.

**Abstract:**

This review highlights recent advances in renal cell carcinoma (RCC) imaging. It begins with dual-energy computed tomography (DECT), which has demonstrated a high diagnostic accuracy in the evaluation of renal masses. Several studies have suggested the potential benefits of iodine quantification, particularly for distinguishing low-attenuation, true enhancing solid masses from hyperdense cysts. By determining whether or not a renal mass is present, DECT could avoid the need for additional imaging studies, thereby reducing healthcare costs. DECT can also provide virtual unenhanced images, helping to reduce radiation exposure. The review then provides an update focusing on the advantages of multiparametric magnetic resonance (MR) imaging performance in the histological subtyping of RCC and in the differentiation of benign from malignant renal masses. A proposed standardized stepwise reading of images helps to identify clear cell RCC and papillary RCC with a high accuracy. Contrast-enhanced ultrasound may represent a promising diagnostic tool for the characterization of solid and cystic renal masses. Several combined pharmaceutical imaging strategies using both sestamibi and PSMA offer new opportunities in the diagnosis and staging of RCC, but their role in risk stratification needs to be evaluated. Although radiomics and tumor texture analysis are hampered by poor reproducibility and need standardization, they show promise in identifying new biomarkers for predicting tumor histology, clinical outcomes, overall survival, and the response to therapy. They have a wide range of potential applications but are still in the research phase. Artificial intelligence (AI) has shown encouraging results in tumor classification, grade, and prognosis. It is expected to play an important role in assessing the treatment response and advancing personalized medicine. The review then focuses on recently updated algorithms and guidelines. The Bosniak classification version 2019 incorporates MRI, precisely defines previously vague imaging terms, and allows a greater proportion of masses to be placed in lower-risk classes. Recent studies have reported an improved specificity of the higher-risk categories and better inter-reader agreement. The clear cell likelihood score, which adds standardization to the characterization of solid renal masses on MRI, has been validated in recent studies with high interobserver agreement. Finally, the review discusses the key imaging implications of the 2017 AUA guidelines for renal masses and localized renal cancer.

## 1. Introduction

Kidney cancer is the 14th most common cancer worldwide, with more than 434,840 new cases diagnosed and 155,953 deaths in 2022 [1]. Renal cell carcinoma (RCC) accounts for 3.5% of all malignancies in Europe [2] and is the most common solid tumor of the kidney. Its incidence has been increasing until recently [3], primarily due to the increased incidental diagnosis of small renal lesions found during abdominal examinations for a variety of indications. Sixty-seven percent of cases are now diagnosed incidentally [4], resulting in a decreasing trend in tumor size and stage [5]. However, most incidentally discovered renal lesions are small and benign, the majority being renal cysts [6,7], while benign solid renal lesions are rarer and mainly represented by angiomyolipomas and oncytomas. The main goal of imaging is to differentiate RCC from benign disease, although in many cases this may not be possible. In fact, approximately 20% of surgically resected renal masses are reported to be benign [8], resulting in increased healthcare costs and exposing patients to surgical risks.

Radiologists play a key role in the diagnosis, characterization, and staging of RCC. In addition, the pretreatment identification of major histologic subtypes of RCC is important because they have different characteristics and clinical behaviors. Clear cell RCC, the most common subtype, accounts for 65–70% of cases and 94% of metastatic RCC and has a 5-year survival rate of 44–59%, whereas papillary RCC (10–15% of RCC) accounts for 4% of metastatic RCC with a survival rate of 82–92% and chromophobe RCC (5% of RCC) accounts for 2% of metastatic RCC with a survival rate of 78–87% [9,10]. Other malignant RCCs (collecting duct carcinoma, MiT family translocation renal cell carcinoma, tubulocystic carcinoma, etc.) are rare and account for 5 to 6% of cases.

CT is the first choice for the characterization of a renal mass and for staging because of its cost and availability [11]. MRI may also be indicated because it offers the added benefits of no radiation exposure and improved characterization of cystic lesions, lesions smaller than 2 cm, and histologic subtypes of RCC [12,13]. However, compared with CT, MRI is more expensive and time-consuming [14]. Based on the American Urological Association (AUA) guidelines, renal biopsy may be considered in patients with suspected renal hematologic or metastatic involvement and in patients with suspected benign renal masses [15] or before active surveillance of RCC.

In recent years, there have been remarkable advances in imaging technology [16]. Current novel imaging modalities include dual-energy CT (DECT) [17,18], photon-counting detector CT [19], radiomics, and high-resolution multiparametric MRI [20]. Compared to single-energy CT, DECT offers new capabilities and provides access to the iodine concentration within renal lesions. It provides images with improved diagnostic performance and a potential reduction in contrast and radiation doses. In addition, the recent introduction of photon-counting detector CT into clinical practice may dramatically change the imaging management of renal masses in the coming years. The introduction of the multi-step interpretation of multiparametric MRI for preoperative assessment of histologic subtypes of RCC and differentiation from benign lesions is also a major advance [20]. Recently, radiomics has been proposed to allow the in-depth assessment of tumor heterogeneity to facilitate precision medicine and better decision making [21]. In addition, due to recent developments, artificial intelligence (AI) has been adopted in the field of radiology and appears to be a useful tool for physicians to make more accurate diagnoses in less time. 

Therefore, the purpose of this review is to provide an overview of the most promising novel imaging approaches in RCC diagnosis.

## 2. Novel Imaging Techniques

### 2.1. Dual-Energy CT

CT is currently the most widely used modality for the initial diagnosis and staging of renal cell carcinoma [11,14]. Over the past decade, there has been an increasing trend toward the use of DECT for the evaluation of renal lesions [17,22,23,24]. Several technical approaches are currently available, including dual-source DECT, single-source rapid kilovoltage switching (fast kVp-switch), single-source sequential (“rotate-rotate”), single-source dual-beam, single-source sequential, and dual-layer spectral multidetector CT. They offer different spectral contrast and dose efficiencies and different post-processing algorithms [22]. 

DECT systems allow essentially simultaneous acquisition of dual-energy images, typically acquired at 80 and 140 kVp, without a significant increase in radiation dose [16]. By acquiring images of the same object at different energies (typically 80 kVp and 140 kVp), DECT is able not only to reconstruct the anatomical structure of the imaged object (conventional CT), but also to approximate the composition of an element contained in the object (spectral CT). Each material has its own spectral response (variation in absorption coefficient) as a function of energy. Thus, two materials with close linear absorption coefficients in one energy band of the radiological spectrum can be completely distinguished from each other by performing measurements in two energy bands.

Post-processing algorithms play a crucial role in DECT. They generate several sets of images: -Monoenergetic images. These images are produced at specific energy levels (e.g., 80 keV, 100 keV, and 140 keV). They provide different levels of contrast and are useful for specific diagnostic tasks, offering improved contrast and tissue visualization compared to conventional polychromatic images.-Optimum contrast images. These result from the non-linear mixing of low-energy images, which enhance contrast, and high-energy images, which provide low noise.

DECT also facilitates the reduction in metal artifacts by post-processing the acquired data.

Differentiation algorithms facilitate the isolation or distinction of specific materials within the data set, often through color coding. Image post-processing algorithms can generate synthetic virtual unenhanced (VUE) images by removing the iodine signal from contrast-enhanced scans, reducing the need for additional scans and minimizing the radiation dose. In a series of 221 patients with 273 renal masses, the differences in renal mass attenuation between VUE and true unenhanced images were within 3 HU for enhancing masses (95% limits of agreement −3.1 HU to 2.7 HU) and non-enhancing cysts (95% limits of agreement −2.9 HU to 2.5 HU) [25]. In addition, the elimination of true enhanced acquisition would result in an estimated mean radiation dose savings of 24% (range 10–36%) for CT renal mass examinations [25]. A large, recently published retrospective study [26] confirmed that there is a strong agreement between VUE and true unenhanced images in the assessment of renal masses. This is important because unenhanced images provide essential information for the classification of renal cystic lesions and the detection of macroscopic fat, hemorrhage, and calcifications and serve as a baseline for comparison with contrast-enhanced images. Numerous studies have shown that CT numbers of virtual unenhanced images are reproducible and comparable to true non-contrast images, allowing the reliable assessment of precontrast renal lesion attenuation [26,27,28]. Nevertheless, Graser et al. [28] observed a difference in attenuation of ≥5 Hounsfield units (HU) between virtual and true unenhanced images in approximately 20% of patients in one study. In addition, significant interscanner variation in attenuation measurements and qualitative assessment of VUE images has been reported [29], particularly in patients scanned on different dual-energy CT scanner types during follow-up imaging. Chandarana et al. [30] also found variability in renal lesion attenuation between virtual and true unenhanced images. Given these findings, it cannot be definitively concluded that VUE images can replace true non-contrast scans. In fact, the effects of reconstruction algorithms, noise reduction techniques, and convolution kernels on the attenuation values of virtual unenhanced and weighted average data remain incompletely understood.

Along with unenhanced images, enhancement on multiphasic CT provides a simple, noninvasive means of suggesting the histologic type of a renal mass. It is defined by an increase of 20 HU or more between precontrast and contrast-enhanced images [11]. In daily practice, an enhancement of <10 HU is considered to be characteristic of a cyst, 10–19 HU of an indeterminate mass, and >20 HU suggestive of a renal tumor. Young et al. [9] showed that the mean enhancement of clear cell RCC (Figure 1) was significantly greater than that of oncocytoma (Figure 2) and chromophobe RCC (Figure 3) in the cortico-medullary and excretory phases, and significantly greater than that of papillary RCC (Figure 4) in the cortico-medullary, nephrographic, and excretory phases. In their series, the mean attenuation values during the corticomedullary phase were 125.0 HU for RCCs, 106.0 HU for oncocytomas, 53.6 HU for papillary RCCs, and 73.8 HU for chromophobe RCCs. However, this quantitative information does not necessarily translate into clinically meaningful measures in daily practice due to the variability and overlap in HU measurements. In a recently published study of 87 patients with 93 pathologically proven papillary RCCs [31], most papillary RCCs presented as a hypovascular, circumscribed, solid renal mass; a few (17%) papillary RCCs presented as the newly defined Bosniak class IIF subtype.

DECT offers several potential advantages over conventional CT in the evaluation of renal masses. 

It may eliminate the unwanted effects of pseudoenhancement by the improved correction of beam-hardening artifacts related to iodine [17].Color-coded iodine overlay images can provide advantages over conventional grayscale imaging for assessing enhancement in subcentimeter lesions, as well as for eliminating the potential errors in region of interest (ROI) positioning for renal tumors that are isodense with the renal parenchyma on the unenhanced image.DECT offers the possibility to obtain a direct quantification of the iodine concentration (in mg/mL) in a lesion, which represents a new option for the characterization of renal masses with equivocal enhancement, especially those with an attenuation baseline between 20 and 70 HU on unenhanced images [32,33]. They may represent either hyperdense cysts or hypovascular true enhancing tumors such as papillary RCC. In these patients, iodine quantification provides a more direct estimate of tumor blood supply and neoangiogenesis.A new area of research has emerged with the implementation of dual-energy maps in discriminating among RCC histologic subtypes [17,18].Quantification of iodine concentration is also of interest for re-evaluation after treatment. Dual-energy iodine quantification could be adopted as an imaging biomarker of tumor viability in cases of advanced RCC treated with targeted or antiangiogenic therapies that reduce tumor perfusion with a limited effect on tumor size [17,30,34].

A meta-analysis reported that DECT had a pooled sensitivity and specificity greater than 95% for evaluation of renal masses, but the accuracy was comparable to that of conventional CT [23]. The authors concluded that larger, multi-institutional studies are needed if DECT is to replace conventional CT in the evaluation of renal masses. A recent single-center study showed a higher confidence in lesion characterization with DECT, with fewer recommendations for additional and follow-up imaging tests than dual-phase single-energy CT and similar performance to MRI [24]. Although the role of DECT to characterize renal masses is growing, the best method for the incorporation of DECT into a renal CT protocol remains to be determined [17,22,23]. 

### 2.2. Photon-Counting Detector CT 

In recent years, the majority of CT instrument development has focused on photon-counting detectors for multispectral CT. Photon-counting detector CT (PCD-CT) is an emerging technology that offers new possibilities for quantification [19]. 

The principle of photon-counting CT (PCD-CT) is based on the use of novel energy-resolving X-ray detectors with mechanisms that differ significantly from those of conventional energy-integrating detectors (EIDs). The novel energy-resolving detectors use semiconducting materials, such as cadmium telluride or cadmium zinc telluride, which allow the direct conversion of X-ray photons into electrical signals. As a result, each photon that interacts with the detector can be individually quantified, allowing precise and individual measurement of its energy. In contrast, conventional CT scanners typically rely on EIDs, which record the total energy deposited in a pixel over a given time period, typically from a large number of photons along with electronic noise. As a result, EIDs only capture photon intensity, while PCDs also capture spectral information, allowing for a more accurate assessment of tissue attenuation properties.

The potential benefits of using a PCD instead of an EID in CT imaging include an improved signal-to-noise ratio, reduced radiation exposure to the patient, improved spatial resolution, mitigation of beam hardening artifacts, and the ability to differentiate between different contrast agents within a single image by using multiple energy bins [35,36]. Ultimately, the CT numbers obtained from conventional CT scanners are dependent on the acquisition protocol and the properties of the surrounding tissues. In contrast, photon-counting CT imparts precise physical material and/or tissue information to each pixel, allowing for more accurate tissue characterization and the visualization of subtle pathologies that may not be apparent on conventional CT images.

The first clinically approved PCD-CT system was cleared by the Food and Drug Administration (FDA) in September 2021. To date, only a few PCD-CT units (less than 30) have been installed worldwide, and clinical research is dominated by validation studies for the implementation of multispectral CT [37]. To date, no publications have specifically focused on the diagnosis of RCC using PCD-CT. 

### 2.3. Multiparametric MR Imaging

In recent years, multiparametric MRI of the kidney has become a key imaging modality for the detection and characterization of renal masses [13,14,24,36].

The multiparametric MR imaging protocol for the evaluation of renal masses typically includes T2-weighted single-shot fast spin-echo sequences (in the axial, coronal, and/or sagittal planes), chemical shift imaging (T1-weighted two-dimensional Dixon GRE in-phase and out-of-phase images in the axial plane), axial diffusion-weighted imaging (SE EPI DWI) with multiple b-values (b = 0–50, 400–500, 800–1000 s/mm^2^) with an ADC map, and dynamic 3D fat-suppressed T1-weighted sequences before and after gadolinium administration (in the axial plane) at 30 s (corticomedullary phase), 90–100 s (nephrographic phase), 180–210 s, and 5–7 min (excretory phase) [13]. Extracellular gadolinium-based contrast material is given at a dose of 0.1 mL per kilogram of body weight injected at 1–2 mL/s, followed by a 10–20 mL saline flush. Image subtraction may be performed during the contrast-enhanced phases to help detect enhancement in lesions where it is equivocal. To be considered complete, the MR protocol should include an evaluation of both kidneys and the liver. 

Each radiologist should be aware of the usefulness of each sequence and its contribution to the diagnosis and ultimately cross-check the various pieces of information provided to arrive at reliable diagnostic hypotheses. 

The detection of macroscopic fat in a renal mass is essential because, in the absence of calcification, it is almost always characteristic of a classic (fat-rich) angiomyolipoma, the most common solid benign renal mass. The macroscopic fat component shows a loss of signal intensity on T1-weighted fat-suppressed images. Angiomyolipomas are also characterized by the presence of an India ink artifact on opposed-phase T1-weighted images at the junction of the mass and normal renal parenchyma, indicating a fat–water interface. T1-weighted gradient-echo inversion recovery imaging allows the detection of microscopic/intracytoplasmic fat. Microscopic intracellular fat is present in clear cell RCC (Figure 5), resulting in a signal drop on opposed-phase images. A signal drop has also been described in angiomyolipomas, including fat-poor angiomyolipomas (Figure 6). Gadolinium-enhanced T1-weighted three-dimensional fat-suppressed gradient-echo imaging is useful to assess the enhancement pattern in a renal mass. It allows the differentiation of hypervascular masses from hypovascular lesions with late and slow enhancement as seen in papillary RCC (Figure 7). T2-weighted sequences are essential for differentiating cystic renal masses from solid renal masses. The T2 signal intensity of a solid renal mass is also helpful in suggesting certain histologic subtypes of RCC. Both fat-poor angiomyolipomas (Figure 6) and papillary RCCs (Figure 7) have a low signal intensity on T2-weighted images, whereas other renal masses have an intermediate or high signal intensity. Several studies have suggested the potential utility of apparent diffusion coefficient (ADC) values to further characterize a renal mass [38]. Both fatty angiomyolipomas and papillary RCCs have low ADC values. 

Cornelis et al. [12,20] proposed a stepwise reading of images organized as follows: (1) T2w images; (2) dual-shift chemical shift MR images; (3) DWI; (4) wash-in analysis of DCE images; and (5) washout analysis of DCE images. The first key feature is the predominant qualitative signal intensity of the lesion on a non-fat-suppressed T2-weighted sequence relative to the renal parenchyma. AMLs with minimal fat content and papillary RCC have a low SI on T2-weighted images, whereas most other solid tumors appear hyperintense or heterogeneous. In addition, chromophobe RCC often appears as a heterogeneous lesion (Figure 8) with a slightly low T2 SI, which allows differentiation from clear cell RCC or renal oncocytoma [39].

The second step is to analyze dual chemical shift MRI to look for the presence or absence of microscopic fat. Clear cell RCC (which contains intracellular microscopic fat) and fat-poor AML often show signal loss on out-of-phase sequences, whereas this has not been reported for renal oncocytoma (Figure 9). This signal loss may be seen sporadically in chromophobe or papillary RCC, but it is very rare.

Third, the DWI sequence needs to be evaluated. Low ADC is often seen in AML and papillary RCC, while ADC remains heterogeneous but is often high in renal oncocytoma and clear cell RCC. As for T2-weighted imaging, chromophobe RCC has a slightly lower ADC compared to these last two lesions. 

Finally, the analysis of DCE seems to be critical. Clear cell RCC, as well as AML, shows a rapid and intense enhancement after contrast injection in the corticomedullary phase (wash in). In oncocytoma and chromophobe RCC, the peak of enhancement is slightly delayed, but washout is observed in all of these tumor subtypes. In papillary RCC, however, enhancement is typically weak and slow, delayed, and maximal in the late phases. Therefore, washout is considered to be absent in papillary RCC. A quantitative approach was proposed by Cornelis et al. [20] to distinguish oncocytomas from chromophobe RCCs (sensitivity 25%, specificity 100%), whereas oncocytomas could be differentiated from clear cell RCC with high sensitivity (100%) and high specificity (94%). The main characteristics of solid renal tumors on multiparametric MRI are summarized in Table 1. 

Although there is growing interest in the use of multiparametric MRI for the diagnosis of renal masses, its accuracy still needs to be validated in large prospective studies.

### 2.4. Contrast-Enhanced Ultrasound

Kidney cancer is often discovered incidentally during an abdominal ultrasound examination. Ultrasound offers a number of advantages, including a low cost, accessibility, and the absence of ionizing radiation. It allows an initial assessment of the size of the tumor, its possible cystic nature, and its vascularization using Doppler [40,41]. However, it cannot be used to perform an exhaustive assessment of extension or to quantify tumor enhancement; moreover, its detection rate of small renal tumors is lower than that of CT, especially those smaller than 2 cm [42]. Renal contrast-enhanced ultrasound (CEUS) has been developed to enable the development of new functional applications for renal blood flow quantification [43]. Second-generation ultrasound contrast agents consist of gas microbubbles stabilized by a phospholipid shell. Measuring 3 to 7 microns, these microbubbles are small enough to pass through pulmonary capillaries and into the arterial system, yet large enough to remain strictly intravascular. There is no intra-tissue passage. Contrast agents have a short half-life (5 min), are not nephrotoxic, and are cleared via the lungs by exhalation [43]. Dynamic real-time acquisition and the use of a purely intravascular contrast agent are the two main features of CEUS that distinguish it from CT and MRI. The purpose of renal contrast-enhanced ultrasound is to study the dynamics of contrast uptake of a lesion to aid in its characterization (Figure 10). In their recent meta-analysis of CEUS features of clear cell RCC less than 4 cm in diameter, Liu et al. [42] reported that hyperenhancement had a moderate sensitivity (67–89%) and specificity (42–75%), while fast-in contrast and heterogeneous enhancement had high diagnostic abilities (area under the curve (AUC) 0.74–0.84); however, the presence of a pseudocapsule and fast-out contrast had poor diagnostic abilities (AUC < 0.70). In 2022, Barr et al. [44] reported a pooled sensitivity of 98% and a pooled specificity of 78% of CEUS for the characterization of solid renal masses in their meta-analysis including 331 patients and 341 lesions. In addition, the high sensitivity (97%) and PPV (98.2%) of CEUS were reported for the diagnosis of RCC in small indeterminate solid renal masses on CT or MRI, excluding lipid-rich AML and cystic renal masses according to the proposed Bosniak classification 2019 [45]. In this retrospective study, CEUS also had a significantly higher sensitivity/NPV for the diagnosis of malignancy in cystic renal masses compared to CT/MRI. These results confirm the trend previously reported by Park et al. [46], who in a retrospective study of 31 cystic masses of the kidney reported diagnostic accuracies of 74% for CT and 90% for CEUS, which were not statistically different. In their series, CEUS images showed more septa in 10 (32%) lesions, more wall thickening and/or septa in 4 (13%) lesions, and more enhancement in 19 (61%) lesions. A recent meta-analysis of the diagnostic performance of CEUS in the evaluation of small renal masses reported an accuracy of 0.93 (sensitivity of 0.94, PPV of 0.95, specificity of 0.78, and NPV of 0.73) in the detection of malignant masses [47]. Although there are few high-powered randomized trials and the exact role of CEUS in the subtyping of renal cancer remains to be determined, CEUS has been included as “usually appropriate” in the American College of Radiology Appropriateness Criteria for the evaluation of indeterminate renal masses in patients with contraindications to CT/MRI contrast [48]. The use of CEUS, which varies widely from country to country, is currently increasing. Further studies are needed to clarify its role in characterizing renal cancer subtypes and stratifying the risk of malignancy of cystic lesions.

### 2.5. Innovative Nuclear Medicine Techniques

In recent years, innovative nuclear medicine techniques have emerged as potentially valuable tools in the characterization of renal tumors. These include sestamibi SPECT/CT and PMSA PET/CT. 

#### 2.5.1. Tc-99m Sestamibi SPECT/CT

Sestamibi, a radiotracer used primarily in cardiac and parathyroid imaging, has recently received attention for its potential application in the study of renal tumors. Sestamibi SPECT/CT is a diagnostic imaging technique that combines single-photon emission computed tomography (SPECT) with computed tomography (CT) and the injection of Tc-99m sestamibi. Tc-99m sestamibi (sestamibi is short for sesta-methoxyisobutylisonitrile) is a technetium radiopharmaceutical. It is a lipophilic cationic compound that selectively accumulates in mitochondria-rich tissues, including tumors. This property is the basis for its utility in imaging solid tumors, including renal tumors. Upon administration, sestamibi undergoes cellular uptake via active transport mechanisms, resulting in its retention within tumor cells. Because oncocytic cells contain large amounts of mitochondria, oncocytomas avidly take up the Tc-99m sestamibi radiotracer. In contrast, most RCCs, especially the clear cell subtype, contain very few or no mitochondria. In addition, many RCCs express multidrug-resistant pumps that transport the radiotracer out of the cells. These two characteristics of RCCs explain why these tumors have a low radiotracer uptake [49]. In 2016, Gorin et al. [50] reported in a prospective trial that Tc-99m sestamibi could differentiate oncocytomas and hybrid oncocytic/chromophobe tumors from RCCs with a sensitivity of 87.5% (95% confidence interval [CI], 47.4–99.7%) and a specificity of 95.2% (95% CI, 83.8–99.4%). There were two falsely positive tumors in their series, and both were of the eosinophilic variant of chromophobe RCC, a rare RCC subtype. A similar diagnostic performance was reported in the meta-analysis of four articles including 117 lesions published by Wilson et al. [51] in 2020. The pooled and weighted sensitivity and specificity of Tc-99m sestamibi SPECT/CT were reported for detecting (1) renal oncocytoma versus other renal lesions at 92% (95% CI 72–98%) and 88% (95% CI 79–94%), respectively; and (2) renal oncocytoma versus chromophobe RCC at 89% and 67%, respectively. All reported studies used a tumor-to-background renal parenchyma radiotracer uptake ratio of >0.6 for positive studies. A cost-effectiveness study showed that Tc-99m sestamibi SPECT/CT followed by confirmatory biopsy could help to avoid surgery for benign small renal masses and minimize untreated malignant small renal masses, and it was cost-effective compared with existing strategies [52]. However this study had data uncertainties and included a limited number of centers from which Tc-99m sestamibi SPECT/CT performance data were collected. Tc-99m sestamibi SPECT/CT is not included in the current recommendations of the EAU guidelines [53], but it appears to be a promising imaging tool that may aid in identifying benign renal oncocytomas and hybrid oncocytic/chromophobe tumors in patients with small renal tumors.

#### 2.5.2. PSMA PET/CT

Initially used in the management of prostate cancer, PSMA (prostate specific membrane antigen) PET/CT has a potential application in suspected RCC. PSMA is a surface receptor antigen expressed in prostate tissue and tumor-associated neovasculature. It is overexpressed in several solid tumors, including RCC. PSMA PET/CT exploits this overexpression to visualize tumor lesions with high specificity. Radiolabeled PSMA ligands, such as 68Ga-PSMA-11 and 18F-DCFPyL, bind to PSMA receptors on tumor cells, enabling the precise imaging of renal tumors. In 2018, an immunochemistry study of 227 RCC patients with a median follow-up of more than 10.0 years showed that the intensity of positive versus negative prostate-specific membrane antigen protein expression was significantly associated with overall survival [54]. There was also a clear trend toward less positive findings in papillary and chromophobe RCC cases. A large immunochemistry study [55] of *n* = 197 papillary RCC type 1 and n = 110 type 2 specimens showed that in papillary RCC type 1, PSMA staining was positive in only 4 of 197 (2.0%) specimens, whereas none (0/110) of the papillary RCC type 2 specimens were positive for PSMA. Clinical studies on the diagnosis of the major subtypes of RCC using PSMA PET/CT are scarce, and its diagnostic performance in this setting has not yet been established. Recent clinical studies have investigated the potential role of PSMA-PET/CT in staging RCC and detecting additional metastatic sites compared to CT or MRI [56,57,58]. They reported a higher sensitivity and PPV of PSMA PET/CT compared to CT, leading to a change in management in 49% of patients in Udovicich’s series [59]. In the latter series, PSMA PET/CT detected additional metastases in 25% of patients compared to CT and had a significantly higher SUVmax than FDG PET/CT [59]. In Aggarwal’s series [60], PSMA PET/CT performed better than CT in detecting bone metastases and worse in detecting liver lesions. Today, current EUA guidelines do not recommend PET in the management of RCC [53], but PSMA PET/CT opens new possibilities that need to be evaluated in large studies.

### 2.6. Radiomics

Radiomics is a rapidly evolving field of research that aims to extract quantitative metrics—so-called radiomic features—from medical images. By mathematically extracting the spatial distribution of signal intensities and pixel interrelationships, radiomics quantifies texture information using analysis methods from the field of artificial intelligence. Texture analysis provides an assessment of tumor heterogeneity by analyzing the distribution and association of pixel or voxel gray levels in the image. Radiomic features capture tissue and lesion characteristics such as heterogeneity and shape and can be used alone or in combination with demographic, histologic, genomic, or proteomic data to improve clinical decision making. It is based on the use of machine-learning algorithms that cannot be perceived by the human eye. The process of radiomics requires several steps, including image acquisition, the segmentation of a volume of interest, and the extraction of features or quantitative values, followed by final correlation with disease staging or clinical data. CT tumor texture analysis is a potentially useful emerging biomarker that has shown promise in predicting tumor histological findings, clinical outcomes [61,62,63,64], overall survival, and the response to therapy. 

The current limitations of radiomics include sensitivity to variations in acquisition parameters (acquisition modes, reconstruction parameters, smoothing and segmentation thresholds) and the limited reproducibility of certain radiomic features. The mannual delineation of tumor volume is not only time-consuming, but may also be affected by inter-observer variability. Further research is needed before radiomics can become part of everyday practice.

### 2.7. Artificial Intelligence

Artificial intelligence (AI), including machine learning and deep learning, has ushered in a new era in the detection and characterization of renal tumors and in clinical decision making [65,66]. Regarding AI and the diagnosis of kidney cancer, several advances have been made in using artificial intelligence to aid in the detection and diagnosis of this disease, as well as in its treatment management. Here are some points to consider:AI can help to detect kidney cancer early by identifying subtle signs or features in medical images that may escape the human eye. This can lead to more effective treatment and improved survival [67].AI has been used to differentiate between benign and malignant tumors [61,68,69,70,71,72].AI has been used to determine the grade and type of malignancy and nuclear atypia of RCC [73].AI can be used to analyze genetic data associated with kidney cancer. By better understanding the genetic profiles of tumors, treatments can be proposed based on the specific characteristics of each patient.Predictive models based on AI can be developed using patient data, including clinical information, imaging data, and test results [74]. These models can help predict the risk of kidney cancer and guide treatment decisions.

## 3. Update of Imaging Algorithms and Guidelines Used for the Diagnosis of RCC

### 3.1. Bosniak Classification of Cystic Masses, Version 2019

Compared with solid masses, cystic renal masses are more likely to be benign and, if malignant, less aggressive [75,76]. Cystic renal cell carcinoma is likely to be overdiagnosed, as suggested by Schoots et al. [76]. In their meta-analysis of 3036 cystic masses, 373 (12%) were malignant and 3 (0.8%) had metastatic disease at presentation, while 49% of Bosniak III cysts were overtreated because of a benign outcome.

The Bosniak classification stratifies the risk of malignancy in cystic renal masses and divides cystic renal masses into five categories based on imaging characteristics on contrast-enhanced CT [75]. It helps to predict the risk of malignancy and suggests either follow-up or treatment. The Bosniak classification should not be used for lesions of infectious, inflammatory, or vascular etiology or in patients with hereditary cystic kidney disease. Since its introduction in 1986, it has been widely used and several refinements have been proposed. The Bosniak classification, version 2019 [75], aims to reduce interobserver variability and improve the categorization and overall accuracy of the imaging-based evaluation of cystic renal masses. The main challenges are to avoid unnecessary surgery for benign lesions and to identify cancers that require treatment. Unlike the original Bosniak classification, which did not include a precise definition of a cystic mass, the new version defines a cystic mass as one that contains less than approximately 25% enhancing tissue. It formally incorporates MRI into the classification system and includes new modality-specific criteria. It also improves the specificity of the higher-risk categories (category III and IV lesions), precisely defines wall/septa and protrusions (Table 2), and improves the clarity of radiologic reporting. It allows a greater proportion of masses to be classified in lower-risk categories. Yan et al. [77] reported improved interreader agreement, while Park et al. [78] reported increased diagnostic specificity for malignancy with the 2019 version. Soon after its release, the Bosniak classification, version 2019, was adopted worldwide, although its widespread validation is still ongoing.

### 3.2. Clear Cell Likehood Score (ccLS)

The clear cell likelihood score (ccLS) is derived from multiparametric MRI with the goal of noninvasively identifying clear cell RCC, the most common subtype of renal cell carcinoma with potentially aggressive behavior. The ccLS was initially developed to address one of the limitations of imaging in renal lesion characterization, namely the lack of standardization in reporting. The first version was released in 2017 followed by a revised version in 2022 [80,81,82]. Automated ccLS calculators are now available online [83]. 

The ccLS system is a standardized framework generated using multiparametric MRI and a guiding algorithm [80,81]. It is based on a step-by-step interpretation of MR images (Table 3) and applies to small solid renal masses without macroscopic fat, less than 4 cm in diameter, with more than 25% of the lesion showing enhancement. It is based on a Likert score of the likelihood of ccRCC; the scoring options include 1 (very unlikely), 2 (unlikely), 3 (intermediate likelihood), 4 (likely), and 5 (very likely). The interpreting radiologist should follow the sequential steps shown in Table 3. They include three types of criteria: (a) eligibility criteria, to ensure that the use of the ccLS is appropriate (as typical angiomyolipomas and cystic renal masses should not be assigned a ccLS); (b) major features, which are mandatory for every renal mass; and (c) ancillary criteria, which are used in specific cases (i.e., to narrow down the differential diagnosis, if directed to do so by the flowchart). In addition, the ccLS can help stratify which patients may or may not benefit from biopsy.

In a recent cohort study of renal masses of any size, Steinberg et al. [84] reported that the PPV for clear cell RCC detection correlated with ccLS (PPV of ccLS1 was 5%, ccLS2 was 6%, ccLS3 was 35%, ccLS4 was 78%, and ccLS5 was 93%). A higher ccLS is also correlated with the faster growth of a small renal mass [85]. 

Integration of ccLS into clinical practice is feasible with few additional requirements beyond the typical standard multiparametric MRI acquisition protocol. It could help to guide patients to surgery, percutaneous renal biopsy, and active surveillance. ccLS use is increasing for both reporting and interdisciplinary discussion. A preliminary CT counterpart to MRI ccLS has recently been proposed [86] but is currently under evaluation.

### 3.3. 2017 AUA Renal Mass and Localized Renal Cancer Guidelines, Renal Mass Biopsy

The 2017 AUA guidelines focus on the evaluation and management of clinically localized sporadic renal masses suspicious for renal cell carcinoma (RCC) in adults, including solid enhancing renal tumors and Bosniak 3 and 4 complex cystic renal masses [14,49]. They focus on management considerations and follow-up of patients with RCC after intervention, but they also include general guidelines for the initial evaluation and diagnosis of RCC. They take into account patient age and comorbidities, the biological and imaging characteristics of the tumor, and renal function considerations (presence of chronic kidney disease (CKD), stage based on glomerular filtration rate (GFR), and degree of proteinuria). 

An AUA guideline states that physicians should obtain high-quality multi-phase cross-sectional abdominal imaging, without specifying whether they should use CT or MRI. MRI offers the added benefits of no radiation exposure, improved characterization of lesions smaller than 2 cm and cystic lesions, and fewer allergic reactions to contrast media. 

Other AUA guidelines state that the description of renal masses should include the size/complexity of the tumor, the presence or absence of macroscopic fat, and the degree of enhancement, with a threshold of 15–20 HU. They remind that “with the exception of fat-containing AML, none of the current imaging modalities can reliably distinguish between benign and malignant tumors or between indolent and aggressive tumor biology”. 

Regarding renal mass biopsy (RMB), the AUA guidelines state that “patients should be counseled regarding the rationale, positive and negative predictive values, potential risks, and nondiagnostic rates of RMB” and expand the usual indications for RMB. They state that clinicians should consider RMB when a mass is suspected to be hematologic, metastatic, inflammatory, or infectious, and that in the setting of a solid renal mass, “RMB should be obtained on a utility-based approach whenever it may influence management”. However, RMB is not necessary in (1) young or healthy patients who are unwilling to accept the uncertainties associated with RMB; or (2) elderly or frail patients who will be managed conservatively regardless of RMB findings. They recommend performing multiple core biopsies, which are preferred over fine needle aspiration (FNA).

## 4. Conclusions

The diagnosis of RCC is evolving. With improved detection due to the increased use of imaging and advances in imaging technology, localized renal cancer accounts for approximately 67% of detected RCC cases. The radiologist plays a pivotal role in the detection, characterization, staging, and subsequent counseling of patients with renal cancer. The updated Bosniak classification, MRI ccLS, and updated AUA Localized Renal Cancer Panel guidelines are now available and should be considered when interpreting imaging studies and during multidisciplinary meetings. New and advanced technologies have also emerged, offering new perspectives for the future. Photon-counting detector CT, radiomics, and AI are the most promising, but further evaluation is needed to determine their exact role in daily practice and personalized medicine. With the further development and improvement of these techniques, there is no doubt that their applications in the imaging diagnosis of RCC will become increasingly widespread. 

## Figures and Tables

**Figure 1 cancers-16-01926-f001:**
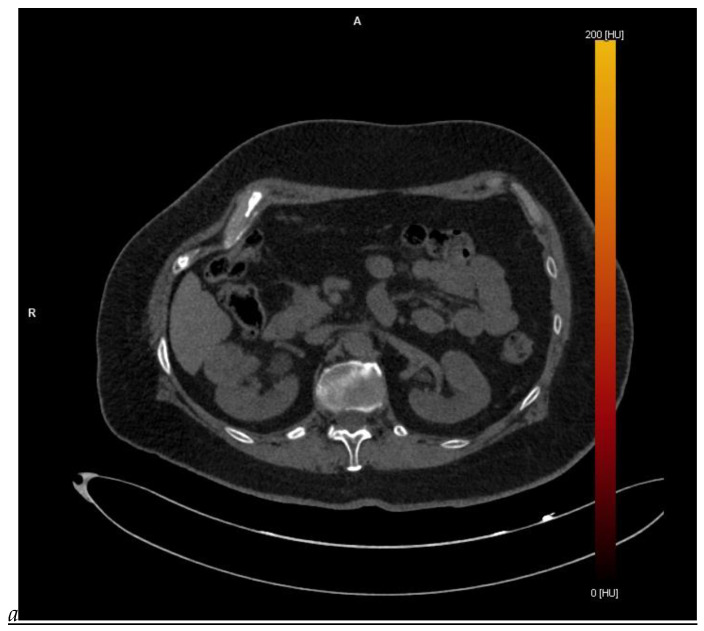
Dual-energy CT aspect of a hypervascularized clear cell renal cell carcinoma of the upper pole of the right kidney. (**a**) Virtual unenhanced image. Note the presence of a solid exophytic renal mass in the upper pole of the right kidney. (**b**) The lesion enhances during the corticomedullary phase; measurement of the iodine concentration of the lesion (4.9 mg/mL) compared to that of the renal cortex (4.2 mg/mL) during the corticomedullary phase. (**c**) Monoenergetic image obtained at 40 keV during the nephrographic phase. Note the decrease in iodine concentration of the lesion (4.42 mg/mL) compared to that of the renal cortex (6.1 mg/mL). (**d**) Monoenergetic image obtained at 70 keV during the excretory phase. Compared to the monoenergetic image at 40 keV, the contrast between the lesion and the adjacent renal cortex is reduced. Note the washout of the lesion (iodine content: 1.8 mg/mL).

**Figure 2 cancers-16-01926-f002:**
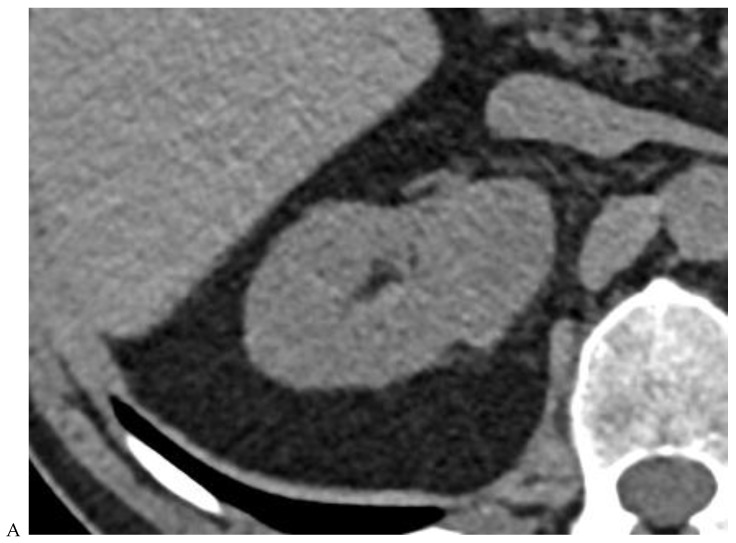
Oncocytoma in the right kidney of a 42-year-old man. (**A**) On the unenhanced image, the 4.8 cm lesion is isodense relative to the renal parenchyma. Enhancement is seen on the corticomedullary phase image (**B**), followed by washout on the nephrographic (**C**) and excretory (**D**) phase images. (**E**) Macroscopic view of the lesion after radical nephrectomy. Courtesy of Pr S. Ferlicot, Department of Pathology, Bicêtre.

**Figure 3 cancers-16-01926-f003:**
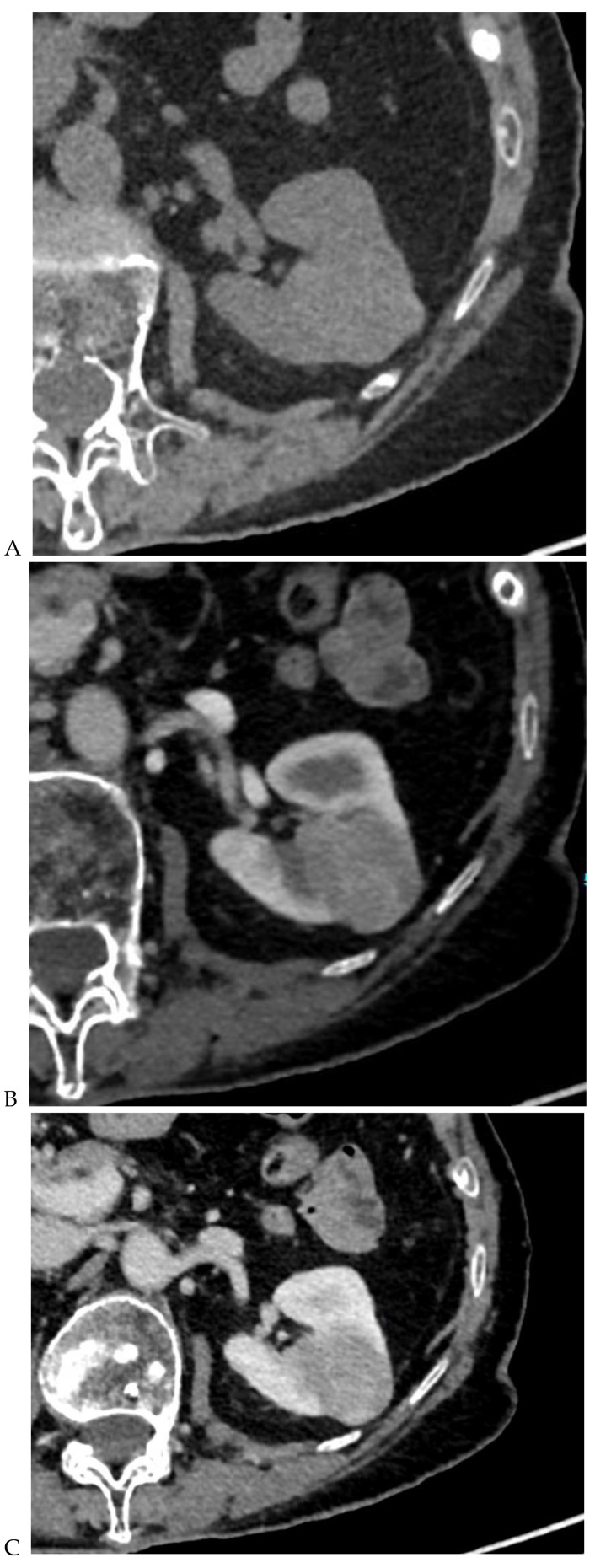
CT aspect of a chromophobe renal cell carcinoma in the left kidney of a 68-year-old-woman. (**A**) Unenhanced image. Presence of an isodense, homogeneous solid lesion at the medium part of the left kidney. (**B**) It appears moderately hypervascularized on the corticomedullary phase image, with hyperdense septa. (**C**) There is progressive washout on the nephrographic phase image and the lesions appears hypodense relative to the renal parenchyma (**C**). (**D**) Macroscopic view of the lesion after partial nephrectomy. Courtesy of Pr S. Ferlicot, Department of Pathology, Bicêtre Hospital.

**Figure 4 cancers-16-01926-f004:**
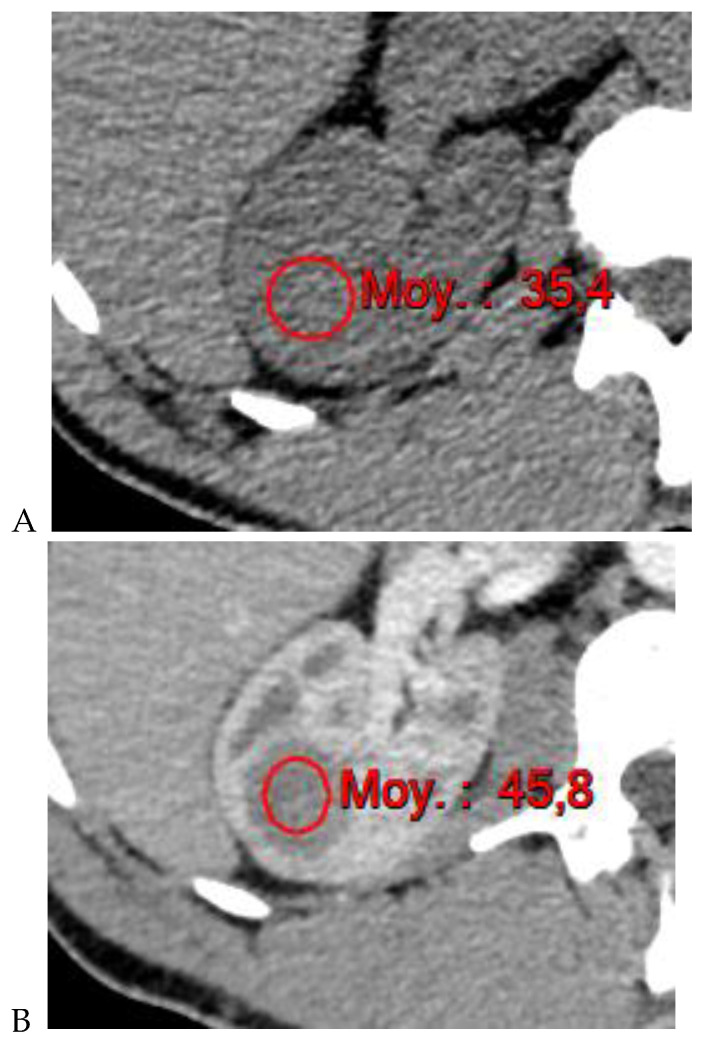
Multiphasic CT enhancement of a papillary renal cell carcinoma in the middle part of the right kidney of a 63-year-old woman. (**A**) Mean unenhanced attenuation was 35 HU. (**B**) Mean corticomedullary phase attenuation was 45 HU. (**C**) Mean nephrographic phase attenuation was 59 HU. (**D**) Mean excretory phase attenuation was 65 HU.

**Figure 5 cancers-16-01926-f005:**
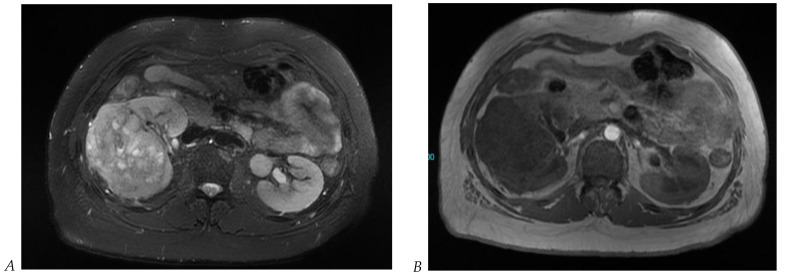
Clear cell renal cell carcinoma in the right kidney of a 52-year-old-man. (**A**) Coronal T2-weighted fast SE image shows a large heterogeneous mass with areas of high signal intensity compared with renal parenchyma. Transverse in-phase (**B**,**C**) opposed-phase MR images show a subtle signal loss on the opposed-phase image. (**D**) The ADC map is heterogeneous with predominant areas of restriction of tumor diffusion. Transverse gadolinium-enhanced T1-weighted gradient-echo spoiled MR images in (**E**) corticomedullary, (**F**) nephrographic, (**G**) and delayed phase images show intense and rapid peripheral enhancement during the arterial and nephrographic phases followed by a rapid washout of contrast on the delayed phase. Central necrotic areas do not enhance. (**H**) Macroscopic view of the lesion after radical nephrectomy. The lesion appears heterogeneous. Courtesy of Pr S. Ferlicot, Department of Pathology, Bicêtre Hospital.

**Figure 6 cancers-16-01926-f006:**
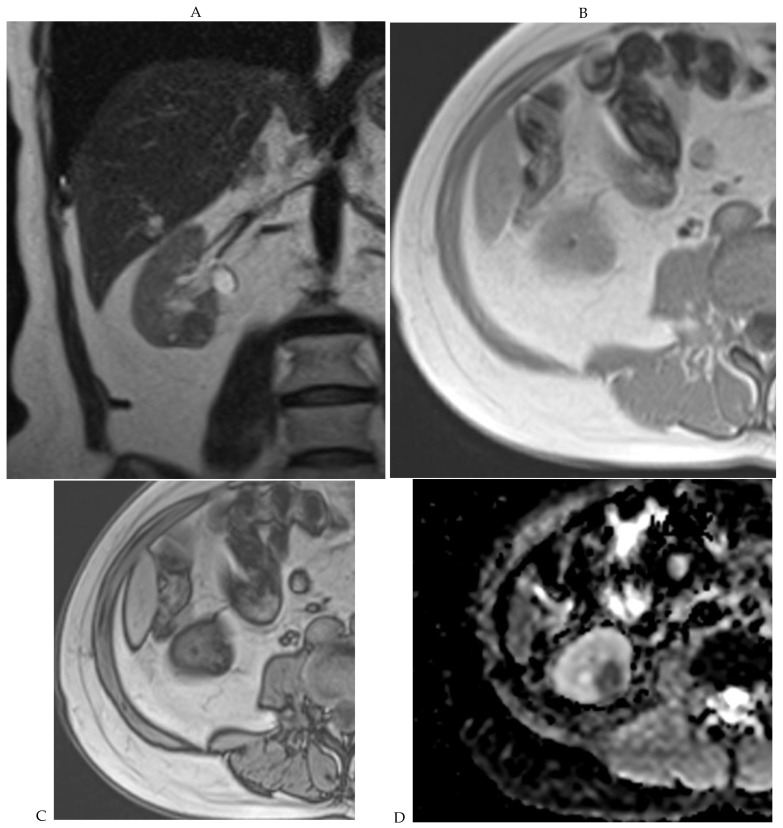
Fat-poor angiomyolipoma in the right kidney of a 46-year-old-man. (**A**) Coronal T2-weighted fast SE image shows the low signal intensity of the lesion compared with the renal parenchyma. Transverse in-phase (**B**,**C**) opposed-phase MR images show a significant loss of signal intensity on the opposed-phase image. (**D**) The ADC map shows marked restriction of tumor diffusion into the renal mass. Transverse gadolinium-enhanced T1-weighted gradient-echo spoiled MR images in (**E**) corticomedullary, (**F**) nephrographic, (**G**) and delayed phase images show early enhancement and rapid washout.

**Figure 7 cancers-16-01926-f007:**
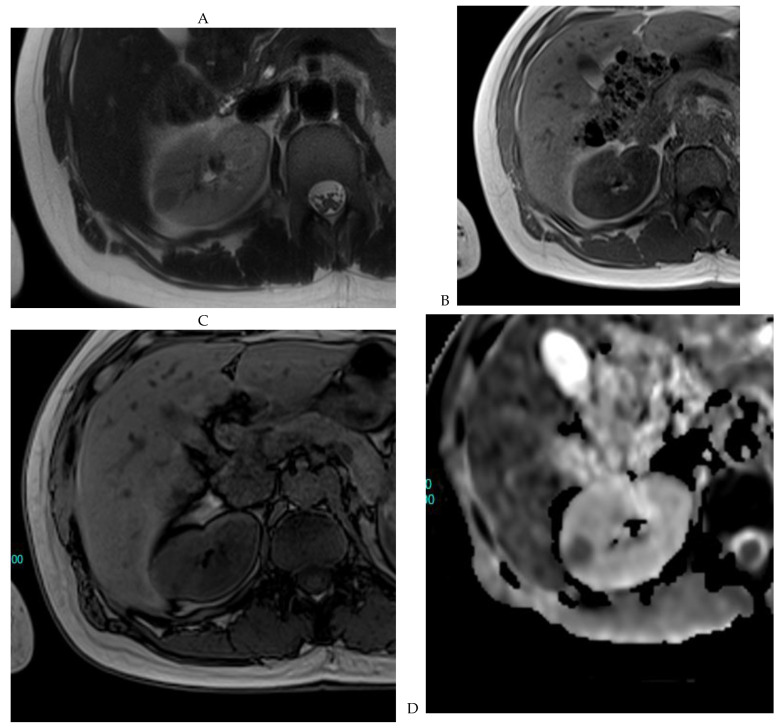
Papillary renal cell carcinoma in the right kidney of a 75-year-old-woman. (**A**) Axial T2-weighted fast SE image shows a homogeneous 1.8 cm mass in the posterolateral region of the right kidney, with a lower SI compared to renal parenchyma. Transverse in-phase (**B**,**C**) opposed-phase MR images do not show a significant signal loss on the opposed-phase image. (**D**) The ADC map shows restriction of tumor diffusion into the renal mass. Transverse nonenhanced (**E**) and gadolinium-enhanced T1-weighted gradient-echo spoiled MR images in (**F**) corticomedullary, (**G**) nephrographic, (**H**) and delayed phase images show progressive enhancement without washout; the mass is hypovascular compared to the renal cortex.

**Figure 8 cancers-16-01926-f008:**
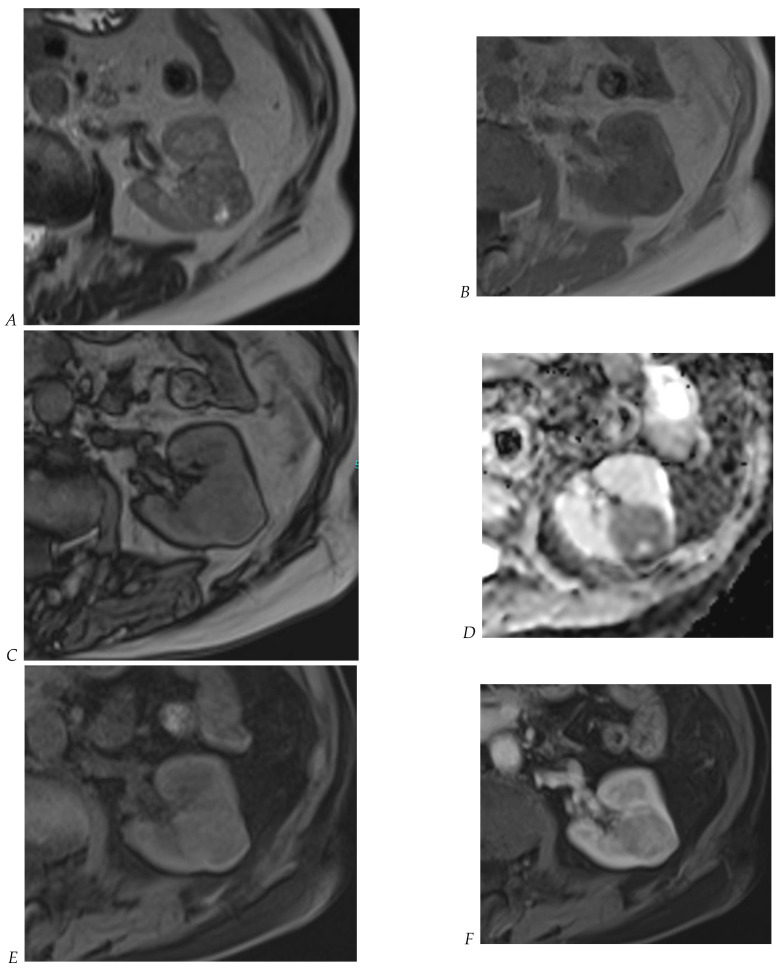
Same lesion as Figure 3. Chromophobe renal cell carcinoma in the left kidney of a 68-year-old-woman. (**A**) Axial T2-weighted fast SE image shows the exophytic heterogeneous isointense renal mass with a posterior hyperintense area. Transverse in-phase (**B**,**C**) opposed-phase MR images show no significant loss of signal intensity on the opposed-phase image. (**D**) The ADC map shows restriction of tumor diffusion into the renal mass. Transverse nonenhanced (**E**) and gadolinium-enhanced T1-weighted gradient-echo spoiled MR images in (**F**) corticomedullary, (**G**) nephrographic, (**H**) and delayed phase images show a mid-intense enhancement of the lesion without visible washout.

**Figure 9 cancers-16-01926-f009:**
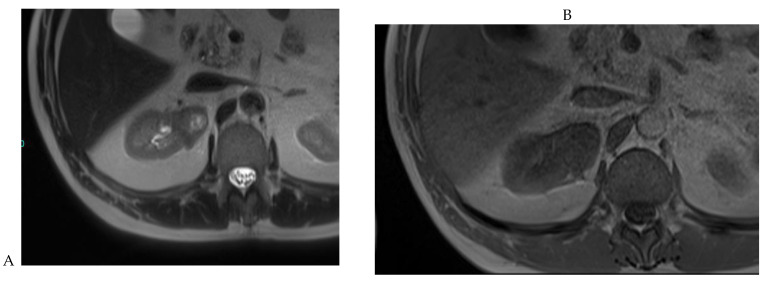
Same lesion as Figure 2. Oncocytoma in the right kidney of a 42-year-old man. (**A**) The axial T2-weighted fast SE image shows a heterogeneous lesion with a central hyperintense area. Transverse in-phase (**B**,**C**) opposed-phase MR images show no significant loss of signal intensity on the opposed-phase image. (**D**) The lesion is hyperintense on the diffusion-weighted image. Transverse nonenhanced (**E**) and gadolinium-enhanced T1-weighted gradient-echo spoiled MR images in (**F**) corticomedullary, (**G**) nephrographic, (**H**) and delayed phase images show early enhancement and rapid washout.

**Figure 10 cancers-16-01926-f010:**
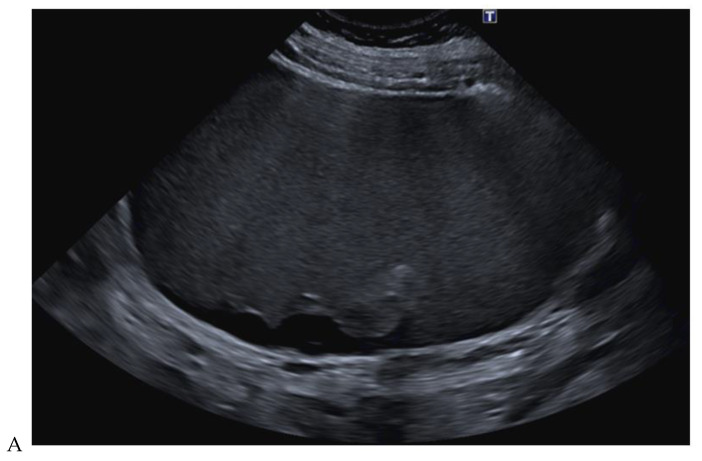
Papillary renal cell carcinoma in the right kidney of a 54-year-old woman. (**A**) B-mode ultrasound shows a large right cystic renal mass with heterogeneous contents and a dependent sediment. (**B**) CEUS with Sonovue^®^ (Bracco Imaging France, Massy, France) reveals the presence of a solid enhancing component at the posterior aspect of the mass. (**C**) The mass was resected. Pathology identified a necrotic papillary renal cell carcinoma. Courtesy of Pr S. Ferlicot, Department of Pathology, Bicêtre Hospital.

**Table 1 cancers-16-01926-t001:** Main characteristics of renal tumors on multiparametric MRI. Adapted from [12,20,39].

Tumor (Sub)Type	T2-Weighted	T1-Weighted	Fat Saturation	Dual Chemical Shift MRI	DCE T1-Weighted	DWI
Angiomyolipoma	Heterogeneous with high SI	Heterogeneous with high SI *	India ink artifact	Signal drop	Arterial enhancement	
Fat-poor angiomyolioma	Low SI			Signal drop	Arterial enhancement	Low
Oncocytoma	Heterogeneous with high SI; central scar			No	Heterogeneous moderate wash-in and washout; late segmental inversion	High
Clear cell RCC	Heterogeneous; central area (necrosis); high SI; pseudocapsule	Heterogeneous high SI of central area		Signal drop	High arterial wash-in and quick washout; heterogeneous	Heterogeneous; high
Papillary RCC	Homogeneous low signal intensity; pseudocapsule			No	Slow and low enhancement	Low
Chromophobe RCC	Heterogeneous central area (necrosis); mid SI			No	Moderate wash-in and washout	Mid

* SI: signal intensity.

**Table 2 cancers-16-01926-t002:** Bosniak classification for CT, version 2019. Adapted from Silvermann SG [75] and Bosniak Classification 2019 [79], https://staging.radiologyassistant.nl (accessed on 2 May 2024).

Bosniak Classification 2019. CT	
Type	Characteristics
I	Well-defined, thin (≤2 mm) smooth wall. Homogeneous fluid (−9 to 20 HU). No septa or calcifications. Wall may enhance.
II	Six types, all homogeneous and well-defined with thin (≤2 mm) smooth walls: - 1. Cystic masses with thin (≤2 mm) and few (1–3) septa. Septa and wall may enhance. May have calcification of any type. - 2. Non-contrast CT: −9 to 20 HU. - 3. Non-contrast CT: ≥70 HU. - 4. Contrast CT: non-enhancing masses, 20 HU at renal mass protocol CT, may have calcification of any type. - 5. Contrast CT: masses: 21–30 HU at portal venous phase. - 6. Homogeneous low-attenuation masses that are too small to characterize.
IIF	Cystic masses with: - 1. Smooth minimally thickened (3 mm) enhancing wall. - 2. Smooth minimal thickening (3 mm) of one or more enhancing septa. - 3. Many (≥4) smooth, thin (≤2 mm) enhancing septa.
III	One or more enhancing thick walls or septa (≥4 mm width). Enhancing irregular walls or septa (displaying ≤ 3 mm obtusely margined convex protrusions).
IV	One or more enhancing nodules (≥4 mm convex protrusion with obtuse margins, or a convex protrusion of any size with acute margins).

**Table 3 cancers-16-01926-t003:** Main steps in determining ccLS (MRI clear cell likehood score).

**Eligibility Criteria**	
- Exclude macroscopic fat	Yes or no
- Confirm > 25% enhancement	Yes or no
**Major features**	
- T2-weighted imaging Signal Intensity on SSFSE sequence	Hypointense, isointense, or hyperintense
- T1-weighted corticomedullary phase enhancement	Mild, moderate, or intense
- Assess presence/absence of microscopic fat	Yes or no
**Ancillary features**	
- Presence of segmental enhancement inversion	Yes or no
- Restriction on DWI images	Yes or no
- Measure arterial-to-delayed enhancement ratio (akin “washout”)	<1.5 or ≥1.5

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
