# Peer review of "Update on Renal Cell Carcinoma Diagnosis with Novel Imaging Approaches"

_cancers, 2024, doi:10.3390/cancers16101926_

Round 1
Reviewer 1 Report
Comments and Suggestions for Authors
I read with great interest the review on Update on Renal Cell Carcinoma Diagnosis with Novel Imaging Approaches.
My suggestion is to add paragraphs to the manuscript.
New imaging approaches are missing, for example, many new papers are exploring the role of SETAMIBI 99 tc. Moreover, the authors should add a paragraph on the role of CEUS. Authors may rely on doi: 10.3390/diagnostics12102310.
A further paragraph on pet-psma and RCC is needed.
Check typos and English revision.
Major revision. Manuscript should be evaluated after these implementations.
Comments on the Quality of English Language
Moderate English revsion
Author Response
Referee 1
We sincerely thank you for your consideration and time spent on the evaluation of our work.
I read with great interest the review on Update on Renal Cell Carcinoma Diagnosis with Novel Imaging Approaches.
Response: We thank you for your encouraging comment.
My suggestion is to add paragraphs to the manuscript.
New imaging approaches are missing, for example, many new papers are exploring the role of SETAMIBI 99 tc. Moreover, the authors should add a paragraph on the role of CEUS. Authors may rely on doi: 10.3390/diagnostics12102310
Response: Thank you for raising these points. A paragraph on Sestamibi SPECT/CT has been added in the revised version. In addition, a paragraph on contrast-enhanced ultrasound has also been added in the revised version. It includes the reference doi: 10.3390/diagnostics12102310
A further paragraph on pet-psma and RCC is needed.
Response: A paragraph on PSMA PET/CT has been added in the revised version
Check typos and English revision.
Response: The English has been checked.
Major revision. Manuscript should be evaluated after these implementations
We hope that all raised questions have been thoroughly addressed, and that the paper will receive a positive evaluation.
Reviewer 2 Report
Comments and Suggestions for Authors
The authors present an interesting study on the current state-of-the-art of RCC imaging. Please find specific comments below:
Abstract:
- Good
Introduction:
- Please add references for the RCC “…accounting for 3-4%...” (i.e. GLOBOCAN)
- Please add reference for “…most incidentally discovered renal lesions are small and benign, the ma- 56 jority being renal cysts, while benign solid renal lesions are rarer and mainly represented 57 by angiomyolipomas and oncytomas.” And add quantification.
- Please quantify high vs. low metastatic potential of RCC subtypes and provide references.
DECT:
- The paragraph L148 – L157 is hard to follow. The authors might want to rephrase and potentially highlight the practicality of enhancement measures, i.e. <10HU no enhancement (cyst), 10-19HU indeterminate, >20HU suggestive of renal tumor.
- Also, the authors might want to provide some quantification of the mean enhancement values for the RCC subtypes and the associated significance. In this context, the authors should also provide how well enhancement can truly differentiate the RCC subtypes (i.e. AUC) – mere HU differences (although statistically significant) might not necessarily translate in clinically meaningful measures if the diagnostic performance is weak (in particular differentiating oncocytoma vs. ccRCC!)
- L162: might want to replace “better” for “improved”
Photon Counting:
- Need to define EID abbreviation
MRI:
- Good
Radiomics:
- Good
AI:
- Additional references might be added at the discretion of the authors, i.e. on discriminating benign vs malignant renal masses using AI vs. radiologists (DOI: 10.1097/MD.0000000000019725) and renal tumor subtypes (DOI: 10.3390/cancers12103010 ), as well as utility in identifying rare renal tumor subtypes (DOI: 10.1007/s00330-021-08201-4)
Updates:
- Please add references and quantification for L329-330
- For table 2, more details are needed want constitutes a point on the ccLS (i.e. is isointense SI on SSFSE one point on the Scale?)
- Also, inclusion vs. exclusion criteria for table 2 need to be stated more clearly
Author Response
Referee 2
We sincerely thank you for your consideration and time spent on the evaluation of our work.
The authors present an interesting study on the current state-of-the-art of RCC imaging.
Response: Thank you for your kind comment.
Please find specific comments below:
Abstract:
- Good
Introduction:
- Please add references for the RCC “…accounting for 3-4%...” (i.e. GLOBOCAN)
Response: Thank you for your comment. The sentence has been modified as follows: “Kidney cancer is the 14th most common cancer worldwide, with more than 434,840 new cases diagnosed and 155,953 deaths in 2022 [Global cancer observatory]. Renal cell carcinoma (RCC) accounts for 3.5% of all malignancies in Europe [Ferlay J] and is the most common solid tumor of the kidney. Its incidence has been increasing until recently [Huang], primarily due to the increased incidental diagnosis of small renal lesions found during abdominal examinations for a variety of indications. » In summary, 3 references have been added, all based on GLOBOCAN.
- Please add reference for “…most incidentally discovered renal lesions are small and benign, the ma- 56 jority being renal cysts, while benign solid renal lesions are rarer and mainly represented 57 by angiomyolipomas and oncytomas.” And add quantification.
Response: Thank you for this comment. Two references have been added:
- Herts BR, Silverman SG, Hindman NM, Uzzo RG, Hartman RP, Israel GM, Baumgarten DA, Berland LL, Pandharipande PV. Management of the Incidental Renal Mass on CT: A White Paper of the ACR Incidental Findings Committee. J Am Coll Radiol. 2018,15, 264-273.
- Corwin MT, Hansra SS, Loehfelm TW, Lamba R, Fananapazir G.Prevalence of solid tumors in incidentally detected homogeneous renal masses measuring > 20 HU on portal venous phase CT. AJR Am J Roentgenol. 2018, 211, W173-W177.
It is difficult to quantitatively define each lesion because their incidence can vary widely according to age, clinical features and imaging modalities.
- Please quantify high vs. low metastatic potential of RCC subtypes and provide references.
Response: Following your comment, we have modified the sentence as follows: “Clear cell RCC, the most common subtype, accounts for 65-70% of cases and 94% of metastatic RCC and has a 5-year survival rate of 44%-59%, whereas papillary RCC (10-15% of RCC) accounts for 4% of metastatic RCC with a survival rate of 82%-92% and chromophobe RCC (5% of RCC) accounts for 2% of metastatic RCC with a survival rate of 78%-87% [Young, Cheville]. »
DECT:
- The paragraph L148 – L157 is hard to follow. The authors might want to rephrase and potentially highlight the practicality of enhancement measures, i.e. <10HU no enhancement (cyst), 10-19HU indeterminate, >20HU suggestive of renal tumor.
Response: Thanks to your suggestion, the sentence has been rephrased as follows: « In daily practice, an enhancement of <10 HU is considered to be characteristic of a cyst, 10-19 HU of an indeterminate mass, and >20 HU suggestive of a renal tumour. »
- Also, the authors might want to provide some quantification of the mean enhancement values for the RCC subtypes and the associated significance. In this context, the authors should also provide how well enhancement can truly differentiate the RCC subtypes (i.e. AUC) – mere HU differences (although statistically significant) might not necessarily translate in clinically meaningful measures if the diagnostic performance is weak (in particular differentiating oncocytoma vs. ccRCC!)
Response: We thank reviewer 2 for this comment. We have added information on HU differences as follows: “In their series, the mean attenuation values during the corticomedullary phase were 125.0 HU for RCCs, 106.0 HU for oncocytomas, 53.6 HU for papillary RCCs, and 73.8 HU for chromophobe RCCs. However, this quantitative information does not necessarily translate into clinically meaningful measures in daily practice due to the variability and overlap of HU measurements.»
- L162: might want to replace “better” for “improved”
Response: “Better” has been replaced by “improved” in the revised manuscript.
Photon Counting:
- Need to define EID abbreviation
Response: Thank you for raising this point. The sentence has been clarified and EID has been defined as energy-integrating detectors: “The principle of photon-counting CT (PCD-CT) is based on the use of novel energy-resolving X-ray detectors with mechanisms that differ significantly from those of conventional energy-integrating detectors (EIDs)”.
MRI:
- Good
Radiomics:
- Good
AI:
Additional references might be added at the discretion of the authors, i.e. on discriminating benign vs malignant renal masses using AI vs. radiologists (DOI: 10.1097/MD.0000000000019725) and renal tumor subtypes : Discriminating malignant and benign clinical T1 renal masses on computed tomography: A pragmatic radiomics and machine learning approach
(DOI: 10.3390/cancers12103010 ), as well as utility in identifying rare renal tumor subtypes
Radiomic Features and Machine Learning for the Discrimination of Renal Tumor Histological Subtypes: A Pragmatic Study Using Clinical-Routine Computed Tomography
(DOI: 10.1007/s00330-021-08201-4) Primary renal sarcomas: imaging features and discrimination from non-sarcoma renal tumors
Response: We express our gratitude to Reviewer 2 for this comment. All 3 suggested references have been added in the revised manuscript. They are references 61, 62, and 72.
Updates:
- Please add references and quantification for L329-330
Response: We appreciate the feedback from referee 2. We have added 2 references (from Silverman SG et al., and from Schoots IG et al.) and incorporated the following sentence to add quantification:
“Compared with solid masses, cystic renal masses are more likely to be benign and, if malignant, less aggressive [Silverman, Schoots]. Cystic renal cell carcinoma is likely to be overdiagnosed, as suggested by Schoots et al. [Schoots]. In their meta-analysis of 3036 cystic masses, 373 (12%) were malignant and three (0.8%) had metastatic disease at presentation, while 49% of Bosniak III cysts were overtreated because of a benign outcome.”
- For table 2, more details are needed want constitutes a point on the ccLS (i.e. is isointense SI on SSFSE one point on the Scale?)
- Also, inclusion vs. exclusion criteria for table 2 need to be stated more clearly
Response: Thank you for raising these points. We have clarified the paragraph as follows:
Automated ccLS calculators are now available online [83].
The ccLS system is a standardized framework generated using multiparametric MRI and a guiding algorithm [80,81]. It is based on a step-by-step interpretation of MR images (Table 3) and applies to small solid renal masses without macroscopic fat, less than 4 cm in diameter, with more than 25% of the lesion showing enhancement. It is based on a Lickert score of the likelihood of ccRCC; the scoring options include 1 (very unlikely), 2 (unlikely), 3 (intermediate likelihood), 4 (likely), and 5 (very likely). The interpreting radiologist should follow the sequential steps shown in Table 3. They include 3 types of criteria: (a) eligibility criteria, to ensure that the use of the ccLS is appropriate (as typical angiomyolipomas and cystic renal masses should not be assigned a ccLS); (b) major features, which are mandatory for every renal mass; and (c) ancillary criteria, which are used in specific cases (i.e., to narrow down the differential diagnosis, if directed to do so by the flowchart). In addition, the ccLS can help stratify which patients may or may not benefit from biopsy.
For your information, please find below a diagram from Ivan Pedrosa’s paper (Radiology 2022) showing the complete algorithm (please see attached file).
We hope that all raised questions have been thoroughly addressed, and that the paper will receive a positive evaluation.
Round 2
Reviewer 1 Report
Comments and Suggestions for Authors
The authors addressed to my major concerns. I endorse pubblication.